# Unraveling the complexity of nurses' quality of life: A Fuzzy-Set Qualitative Comparative Analysis (fsQCA) approach to the interplay of psychological, occupational, and social factors

Xu Jun[1], Li Shasha[2], Mao Dandan[1]*

**1** Ningbo Municipal Hospital of Traditional Chinese Medicine (TCM), Affiliated Hospital of Zhejiang Chinese Medical University, China, **2** department of medicine, Huzhou University, Huzhou, Zhejiang Province, China

☯ Li Shasha is the co-first author.

* jessicamdd@163.com

## Abstract

### Aim

This study aimed to explore the interplay among occupational stress, psychological distress, social support, burnout, and work-family conflict in relation to the quality of life (QOL) of nurses.

### Methods

A total of 503 nurses from two tertiary-level hospitals participated in an online survey. Data were collected on psychological distress, occupational stress, social support, burnout, work-family conflict, and quality of life. Fuzzy-set Qualitative Comparative Analysis (fsQCA) was conducted to identify configuration pathways associated with high and low QOL, thereby providing a nuanced understanding of the causal complexity underlying these outcomes.

### Results

Four configuration pathways were identified as conducive to high QOL: (H1) low burnout, low occupational stress, and high social support; (H2) low work-family conflict, low burnout, low occupational stress, and low psychological distress; (H3) high social support, low work-family conflict, low occupational stress, and low psychological distress; and (H4) high social support, low work-family conflict, low burnout, and low psychological distress. Conversely, two pathways were associated with low QOL: (L1) high occupational stress, high psychological distress, high work-family conflict, and low social support; and (L2) high occupational stress, high psychological distress, high work-family conflict, and high burnout.

**Data availability statement:** Data cannot be shared publicly because of participant privacy protections and institutional data governance regulations. Data are available from the Ningbo Municipal Hospital of Traditional Chinese Medicine (TCM), Affiliated Hospital of Zhejiang Chinese Medical University Institutional Data Access / Ethics Committee (contact via nbhospital@126.com]) for researchers who meet the criteria for access to confidential data.

**Funding:** The author(s) received no specific funding for this work.

**Competing interests:** The authors have declared that no competing interests exist.

## Conclusion

Nurses' QOL is critically influenced by occupational and psychological stressors, work-family conflict, and social support. Addressing these factors through targeted interventions may reduce burnout and psychological distress while enhancing well-being and job satisfaction.

---

## 1. Introduction

The quality of life (QOL) of nurses has emerged as a critical concern within the global healthcare system, significantly impacting the well-being of healthcare professionals, patient outcomes, and the overall efficiency of healthcare delivery. QOL is a multidimensional construct that encompasses physical, psychological, social, and environmental dimensions, reflecting an individual's overall life satisfaction and functional capacity [1]. Studies have consistently shown that nurses report lower QOL compared to the general population, primarily due to high levels of occupational stress, psychological distress, burnout, and work-family conflict [2,3,4,5,6,7]. For example, factors such as extended working hours, emotional demands, and resource limitations contribute to chronic stress and emotional exhaustion, which not only compromise nurses' QOL but also adversely affect their ability to provide high-quality care [8,9]. A qualitative study reveals that nurses' quality of life is significantly undermined by workload pressures and other systemic factors, which contribute to dissatisfaction and eventual resignation [10]. Moreover, diminished QOL among nurses has been linked to adverse patient outcomes, including higher rates of medical errors, patient dissatisfaction, and compromised safety [11,12,13]. The repercussions of poor QOL extend beyond individual nurses, affecting team cohesion, institutional reputation, and healthcare efficiency [14]. Addressing the factors that influence nurses' QOL is crucial for building a resilient nursing workforce and ensuring sustainable healthcare delivery.

Studies indicate that nurses' QOL is influenced by a combination of internal and external factors, such as occupational stress, psychological distress, burnout, social support and work-family conflict. Occupational stress is an unavoidable issue in nursing, characterized by heavy workloads, long working hours, and the psychological demands of caring for critically ill patients [15]. Prolonged exposure to high-pressure environments depletes personal resources, leading to physical and mental health deterioration, ultimately reducing QOL [16,7]. Psychological distress, defined as a range of negative mental states such as anxiety, depression, and emotional exhaustion, also has a significant impact on nurses' QOL [17,18]. Nurses exposed to high-intensity and high-risk work environments are prone to accumulating chronic stress, which adversely affects their mental health. Kchaou et al. found that nurses experiencing high levels of psychological distress report significantly lower life satisfaction, a state that further diminishes their overall QOL [19]. Burnout is another prevalent issue among nurses, profoundly affecting their QOL. Characterized by emotional exhaustion, depersonalization, and reduced personal accomplishment, burnout

erodes nurses' professional passion and life satisfaction [20]. A systematic review found that nurses with higher levels of burnout are more likely to experience psychological exhaustion, negatively impacting their QOL [3]. Work-family conflict also plays a crucial role in influencing nurses' QOL. According to the Work-Family Conflict Theory, competing demands between work and family create resource allocation challenges, making it difficult for individuals to meet both responsibilities, thereby generating psychological stress [21]. Dilmaghani observed that high levels of work-family conflict significantly increase nurses' psychological distress and burnout, further reducing their QOL [22]. Conversely, social support is widely recognized as a critical factor in improving QOL [23]. Shojaei et al highlighted that support from family, colleagues, and management can help nurses manage stress, improve mental and physical health, and significantly enhance their overall QOL [24].

To provide a coherent theoretical basis for understanding these relationships, this study draws on two complementary frameworks. The Conservation of Resources (COR) Theory posits that individuals strive to acquire and protect valued resources—including energy, social relationships, and professional competence—and that stress arises when resources are threatened, lost, or fail to accrue following investment [25]. In the nursing context, high occupational demands and work-family conflict accelerate resource depletion, while social support serves as a key resource buffer. Complementarily, the Job Demands-Resources (JD-R) Model distinguishes between job demands that drive exhaustion and burnout, and job resources that foster engagement and well-being [26]. Together, these frameworks suggest that nurses' QOL is shaped not by individual stressors or resources in isolation, but by the interplay between demands and resources—precisely the kind of configurational complexity that fsQCA is designed to capture.

Although existing studies have extensively explored the multidimensional factors influencing nurses' QOL, there is still a lack of precise and efficient interventions to address nurses' QOL. The primary challenge lies in the multidimensional and intricate nature of the factors influencing nurse QOL. Prior studies have mainly relied on logistic regression or structural equation modeling to explore these factors. Although these approaches have enhanced our understanding, they are not without limitations. Specifically, their reductionist and linear methodologies focus on the isolated effects of individual variables, often overlooking the systemic and holistic aspects that are integral to nurse QOL [24]. Nurse QOL arises from a complex interplay of multiple factors, therefore, a comprehensive examination of these interactions is essential. While regression analysis can assess interactions between variables, it faces challenges when dealing with interactions among three or more variables, such as multicollinearity and model bias [27]. In contrast, Qualitative Comparative Analysis (QCA), which uses Boolean algebra to identify patterns among causal conditions across cases, provides a more holistic approach [24]. This method acknowledges that variables influence outcomes through their interrelationships rather than in isolation.

Consequently, this study adopts QCA to uncover the causal mechanisms underlying nurse QOL and to determine the differential impacts of each factor. This research is crucial for comprehending the complex causal dynamics of nurse QOL and for developing targeted, cost-effective interventions. By employing QCA, this study aims to provide a deeper understanding of how various factors interact to affect nurse QOL, ultimately contributing to the development of more effective and practical solutions.

## 2. Methods

### 2.1. Design

This study employed a cross-sectional design.

### 2.2. Participants

This study was conducted in Zhejiang Province, China. Both participating hospitals are tertiary-level general hospitals located in urban Ningbo, providing comprehensive medical and specialist services to large and diverse patient

populations. A convenience sampling method was used to recruit 503 clinical nurses from two tertiary general hospitals. Inclusion criteria were: (1) registered nurses actively working during the study; (2) at least one year of clinical nursing experience; and (3) informed consent to participate. Exclusion criteria were: (1) nurses attending external training programs; (2) those on extended leave (e.g., medical or maternity leave); and (3) visiting nurses or nursing interns. Participants were also excluded if their survey response time was under 3 minutes or if their answers showed patterned or inconsistent response styles. A total of 520 questionnaires were distributed electronically. After excluding 17 invalid responses, 503 valid questionnaires were retained, yielding a valid response rate of 96.73%.

## 2.3. Ethics statement

The research protocol was reviewed and approved by the Ethics Committee of Ningbo Municipal Hospital of Traditional Chinese Medicine [KYSL-2024-001-059]. Informed consent was obtained from all individual participants prior to their participation in the study. Participants were provided with comprehensive information regarding the study's objectives, procedures, potential risks and benefits, and measures to ensure confidentiality and data protection. Written informed consent was obtained from all participants, and signed consent forms were securely stored in accordance with institutional guidelines.

## 2.4. Measurements

**2.4.1. *Demographic characteristics and working characteristics*.** The survey included the following variables: age, gender, educational attainment, marital status, presence of children, monthly income, job position, years of work experience, and frequency of night shifts.

**2.4.2. *8-Item Short Form Health Survey, SF-8*.** The 8-Item Short Form (SF-8) was used in this study to measure nurses' QOL. The SF-8 includes eight items across eight dimensions: physical function, role physical, bodily pain, general health, vitality, social function, role emotional, and mental health [28]. Each item is scored on a 5-point Likert scale, with 1 indicating the poorest state and 5 indicating the best state. Higher total scores reflect higher levels of QOL. Studies have demonstrated that the Chinese version of the SF-8 has good reliability and validity [29]. In this study, the Cronbach's α for the scale was 0.91, indicating excellent internal consistency.

**2.4.3. *Kessler Psychological Distress Scale,K10*.** The Kessler Psychological Distress Scale (K10) was used in this study to assess nurses' psychological distress [30]. This scale evaluates non-specific mental health conditions, such as anxiety and depression, experienced over the past four weeks. It comprises 10 items scored on a 5-point Likert scale. Higher scores indicate poorer mental health. The total score ranges from 10 to 50, with scores below 16 indicating minimal distress and scores of 16 or higher indicating mild to severe distress. The Cronbach's α for the scale in this study was 0.90, demonstrating excellent internal consistency.

**2.4.4. *Occupational Role Scale*.** The Occupational Role Scale, revised by Feng et al., was used in this study to measure nurses' occupational stress [31]. The scale consists of 22 items across four dimensions: task overload, task conflict, work responsibility, and work environment. Each item is rated on a 5-point Likert scale. Higher scores indicate higher levels of occupational stress. This scale has been used in clinical nurses [32]. In this study, the Cronbach's α for the scale was 0.87, indicating good internal consistency.

**2.4.5. *Maslach Burnout Inventory 2-item, MBI-2*.** MBI-2 was developed by Li-Sauerwine et al. consists of two items measuring emotional exhaustion and depersonalization. Each item is rated on a 7-point Likert scale, with a total score of 14. Higher scores indicate higher levels of occupational burnout. The MBI-2 has been validated as a reliable tool for the rapid screening of burnout among healthcare workers [33]. In this study, the Cronbach's α for the scale was 0.81.

**2.4.6. *Perceived Social Support Scale, PSSS*.** The PSSS was developed by Zimet et al. [34] in 1990 and later adapted and revised to Chinese version by Jiang [35]. The scale consists of three dimensions: family support, friend

support, and other support, with a total of 12 items. The higher the total score, the greater the perceived level of social support. The total score ranges from 12 to 36, indicating low levels of support; from 37 to 60, indicating moderate levels of support; and from 61 to 80, indicating high levels of support. In this study, the Cronbach's α coefficient for the PSSS was 0.92.

**2.4.7. *Work-Family Conflict Scale*.** The Work-Family Conflict Scale measures the extent of conflict experienced by nurses between their home and work environments [36]. It comprises ten items and uses a 7-point Likert scale, with total scores ranging from 10 to 70. Higher scores indicate more severe work-family conflicts. The scale is noted for its good reliability, with a Cronbach's α of 0.88 in this study.

## 2.5. Data collection

Data were collected from 20 April to 30 May,2025. Data were collected using structured self-report questionnaires administered to nurses from two tertiary general hospitals. Participants were recruited through convenience sampling and were informed about the purpose and procedure of the study before providing informed consent. The questionnaires were distributed electronically via an online survey platform to ensure accessibility and convenience. To enhance data quality, responses were excluded if the completion time was less than 3 minutes or if the answers exhibited patterned or inconsistent response styles. The data collection process lasted for three months, and confidentiality of participants' responses was strictly maintained throughout the study.

## 2.6. Data analyze

Descriptive analysis was conducted using SPSS 26.0. For measurement data following a normal distribution, mean±standard deviation were used for description, while frequency and component ratio were used for counting data. The correlation between variables was examined through Pearson correlation analysis. Independent t-tests and one-way ANOVA were used to compare differences in quality of life across demographic subgroups. To further investigate the causal configurations of factors influencing quality of life, a fuzzy-set Qualitative Comparative Analysis (fsQCA) was performed using fsQCA 3.0 software. FsQCA was used to identify distinct combinations of psychological distress, occupational stress, burnout, and social support that led to high or low quality of life. The consistency and coverage thresholds were set following established guidelines to ensure the robustness of the results [37].

## 3. Results

### 3.1. Descriptive and univariate analysis

A total of 503 nurses participated in the study, with 384 (76.3%) being female and 450 (89.5%) married. Among the participants, 103 nurses (20.5%) reported an income of less than 5000 RMB per month, and 113 (22.5%) had ≤ 5 years of working experience. Additionally, 84 nurses (16.7%) did not work night shifts, and a majority had no research experience. The results revealed that age, marital status, income, working years, and night shifts were influential variables affecting nurses' QOL. The differences in QOL across these variables were all statistically significant ($p < 0.05$), as shown in Table 1.

### 3.2. The relationships between psychological distress, occupational stress, social support, burnout,work-family conflict and QOL

Pearson's correlation analysis showed that psychological distress ($r = -0.343$, $p < 0.01$), occupational stress ($r = -0.643$, $p < 0.01$), burnout ($r = -0.419$, $p < 0.01$), work-family conflict ($r = -0.235$, $p < 0.01$) was negatively correlated with QOL, social support was positively correlated with QOL ($r = 0.615$, $p < 0.01$).

**Table 1. Differences in QOL across demographic characteristics.**

| | Item | N | QOL | X² | P |
|---|---|---|---|---|---|
| Gender | Male | 119 | 24.18±4.03 | 0.568 | 0.451 |
| | Female | 384 | 24.50±4.15 | | |
| Age | <25 | 125 | 22.85±4.04 | 6.937 | <0.001 |
| | 25-30 | 88 | 24.44±3.98 | | |
| | 31-35 | 96 | 25.05±4.05 | | |
| | 36-40 | 98 | 24.94±3.39 | | |
| | >40 | 96 | 25.31±4.62 | | |
| Education level | College degree or below | 142 | 24.36±4.08 | 0.232 | 0.793 |
| | Undergraduate | 193 | 24.32±3.86 | | |
| | Master or above | 168 | 24.60±4.44 | | |
| Marriage status | Unmarried | 135 | 24.66±3.98 | 4.107 | 0.007 |
| | Married | 350 | 24.50±4.12 | | |
| | Divorced | 15 | 21.40±4.40 | | |
| | Widowed | 3 | 20.00±1.73 | | |
| Has child or not | Yes | 246 | 24.22±4.34 | 1.250 | 0.264 |
| | No | 257 | 24.63±3.91 | | |
| Income (per month) | <5000RMB | 103 | 23.17±4.41 | 6.083 | <0.001 |
| | 5000-7000RMB | 111 | 23.86±3.59 | | |
| | 7000-10000RMB | 97 | 24.37±3.66 | | |
| | 10000-12000RMB | 111 | 25.41±4.06 | | |
| | >12000RMB | 81 | 25.49±4.48 | | |
| Management position | Yes | 256 | 24.45±4.53 | 0.023 | 0.878 |
| | No | 247 | 24.40±3.66 | | |
| Working years | ≤5 years | 113 | 24.97±4.43 | 6.014 | <0.001 |
| | 6-10 years | 109 | 25.38±3.76 | | |
| | 11-15 years | 140 | 24.33±3.51 | | |
| | ≥16 years | 141 | 23.35±4.46 | | |
| Night shifts (per month) | 0 | 80 | 25.30±4.48 | 5.496 | <0.001 |
| | 1-4 | 182 | 24.96±4.21 | | |
| | 5-9 | 125 | 24.15±3.82 | | |
| | ≥10 | 116 | 23.28±4.12 | | |

### 3.3. Hierarchical regression model with QOL

Prior to hierarchical regression analysis, all relevant statistical assumptions were verified. Normality of residuals was confirmed via P-P plot inspection. Multicollinearity was assessed using Variance Inflation Factor (VIF) and tolerance statistics; all VIF values were below 5 (range: 1.06–2.05) and all tolerance values exceeded 0.20, indicating no significant multicollinearity among predictors. A variance variation of 0.117 was obtained after entering the sociodemographic factors into the regression model, with significance in the one-way analysis. In the second step, three psychological variables were added, and it was found that a variance variation of 0. 631was obtained. The significant variables were occupational stress ($\beta = -0.374$, $p < 0.01$) and social support ($\beta = 0.328$, $p < 0.01$), as can be seen in Table 2.

**Table 2. Hierarchical regression model with the QOL.**

| | Non-standardized coefficient | | Standardized coefficient | t | p |
|---|---|---|---|---|---|
| | β | SE | β | | |
| **Model 1** | | | | | |
| (constant) | 25.688 | 1.053 | | 24.401 | .000 |
| Age | .477 | .120 | .169 | 3.972 | .033 |
| Marriage status | −.713 | .333 | −.090 | −2.141 | .000 |
| Frequency of night shifts | −.704 | .171 | −.173 | −4.120 | .000 |
| Monthly income | .495 | .128 | .166 | 3.861 | .013 |
| Working years | −.395 | .159 | −.107 | −2.482 | .000 |
| R² | 0.126 | | | | |
| Δ R² | 0.117 | | | | |
| F | 14.325 (P < 0.001) | | | | |
| Model 2 | | | | | |
| (constant) | 33.589 | 1.584 | | 21.212 | .000 |
| Age | .194 | .079 | .069 | 2.461 | .014 |
| Marriage status | −.145 | .217 | −.018 | −.667 | .505 |
| Frequency of night shifts | −.367 | .112 | −.090 | −3.281 | .001 |
| Monthly income | .111 | .085 | .037 | 1.302 | .193 |
| Working years | −.105 | .104 | −.029 | −1.014 | .311 |
| Social support | .167 | .016 | .328 | 10.410 | .000 |
| Work-family conflict | −.040 | .017 | −.066 | −2.367 | .018 |
| Burnout | −.237 | .040 | −.172 | −5.888 | .000 |
| Psychological distress | −.116 | .017 | −.188 | −6.692 | .000 |
| Occupational Stress | −.151 | .013 | −.374 | −11.778 | .000 |
| R² | 0.638 | | | | |
| Δ R² | 0.631 | | | | |
| F | 86.807 (P < 0.001) | | | | |

## 3.4. Necessary analysis

Calibration of fuzzy-set membership scores was performed using the direct method in fsQCA 3.0. For each condition, three anchor points were specified: full membership (0.95), the crossover point (0.50), and full non-membership (0.05). Thresholds were determined based on the theoretical range of each instrument and the empirical distribution of scores in our sample. Seen in Table 3.

The consistency score was used to evaluate whether an antecedent variable was essential for the outcome variable. Similar to the significance of a coefficient in regression, the consistency score reflects the extent to which the outcome depends on the antecedent variable. In this study, none of the antecedent variables reached the threshold of 0.9, indicating that no variable could be identified as a necessary condition. Seen Table 4 for details.

## 3.5. Sufficient analysis

Four conditional configurations for high QOL and two for low QOL were analyzed. These six configurations were identified as sufficient conditions for determining high and low QOL among nurses. The overall solution coverage was 0.667 for high QOL and 0.545 for low QOL, as shown in Table 5. This indicates that the six configurations explain 66.7% of the variance for high QOL and 54.5% for low QOL. The configurations are elaborated as follows: ①H1: low burnout + low occupational stress + high social support. ②H2:low work-family conflict + low burnout + low occupational stress + low psychological

**Table 3. Necessity analysis for QOL.**

| | Full Membership (0.95) | Crossover Point (0.50) | Full Non-Membership (0.05) |
|---|---|---|---|
| Occupational Stress | 58 | 46 | 32 |
| Psychological Distress (K10) | 50 | 38 | 27 |
| Burnout (MBI-2) | 13 | 7 | 3 |
| Work-Family Conflict | 44.9 | 30 | 23 |
| Social Support (PSSS) | 76.9 | 60 | 43 |
| QOL (SF-8) | 31 | 24 | 18 |

**Table 4. Necessity analysis for QOL.**

| Variable | High QOL | | Low QOL | |
|---|---|---|---|---|
| | Consistency | Coverage | Consistency | Coverage |
| Social Support | 0.786 | 0.812 | 0.558 | 0.537 |
| ~ Social Support | 0.552 | 0.573 | 0.805 | 0.778 |
| Work-family Conflict | 0.600 | 0.625 | 0.701 | 0.681 |
| ~ Work-family Conflict | 0.694 | 0.713 | 0.615 | 0.589 |
| Burnout | 0.549 | 0.573 | 0.750 | 0.729 |
| ~Burnout | 0.741 | 0.761 | 0.561 | 0.537 |
| Psychological Distress | 0.618 | 0.619 | 0.756 | 0.706 |
| ~Psychological Distress | 0.706 | 0.757 | 0.592 | 0.591 |
| Occupational stress | 0.557 | 0.540 | 0.845 | 0.763 |
| ~Occupational stress | 0.755 | 0.839 | 0.491 | 0.508 |

**Table 5. Sufficient conditions for the intermediate solution of low & high QOL.**

| | High QOL | | | | Low QOL | |
|---|---|---|---|---|---|---|
| | H1 | H2 | H3 | H4 | L1 | L2 |
| Social Support | ● | | ● | ● | ⊗ | ⊗ |
| Work-family conflict | | ⊗ | ⊗ | ● | ● | |
| Burnout | ⊗ | ⊗ | | ⊗ | | ● |
| Psychological distress | | ⊗ | ⊗ | ⊗ | ● | ● |
| Occupational stress | ⊗ | ⊗ | ⊗ | | ● | ● |
| Raw coverage | 0.544 | 0.396 | 0.425 | 0.356 | 0.454 | 0.474 |
| Unique coverage | 0.116 | 0.023 | 0.052 | 0.048 | 0.071 | 0.090 |
| Consistency | 0.956 | 0.960 | 0.959 | 0.949 | 0.915 | 0.931 |
| Solution coverage | 0.931 | | | | 0.913 | |
| Solution consistency | 0.667 | | | | 0.545 | |

distress. ③H3: high social support +low work-family conflict +low occupational stress +low psychological distress. ④H4: high social support＋low work-family conflict＋low burnout＋low psychological distress. ⑤L1: high occupational stress+high psychological distress+ high work-family conflict+ low social support. ⑥L2:high occupational stress＋high psychological distress＋high work-family conflict＋high burnout.

## 4. Discussion

This study employs fuzzy-set Qualitative Comparative Analysis (fsQCA) to examine the configurations of variables influencing nurses' QOL. It offers a fresh perspective on the interactions and combinations of psychological, occupational, and social factors that shape QOL. The findings deepen understanding of these factors and provide practical insights for designing targeted interventions to enhance nurses' QOL and promote the sustainability of high-quality nursing teams.

Statistical analysis indicates significant associations between nurses' quality of life (QOL) and several factors, such as psychological distress, occupational stress, burnout, work-family conflict, and social support. These results align with earlier studies [22,23,38,39]. Nevertheless, this straightforward correlation prompts an important inquiry: is it necessary to address all these factors concurrently to effectively improve nurses' QOL? Prior research has shown that comprehensive but non-specific interventions frequently lead to inefficacy and wasteful resource use [40]. Utilizing fuzzy-set Qualitative Comparative Analysis (fsQCA), this study uncovers distinct combinations of these factors, providing a more precise and resource-efficient strategy for enhancing nurses' QOL by concentrating on critical configurations instead of individual elements. These configurational findings are consistent with the theoretical frameworks outlined in the introduction. From a COR Theory perspective [Hobfoll, 1989], the low-QOL configurations (L1, L2) represent states of severe resource depletion, where high occupational and psychological demands exhaust personal resources without adequate replenishment through social support. Conversely, the high-QOL configurations (H1–H4) reflect conditions of resource conservation and recovery, where reduced demands and enhanced social support enable nurses to sustain their well-being. The JD-R Model further contextualizes these findings: the identified configurations reveal that it is specifically the imbalance between job demands and available resources—rather than any single factor—that drives QOL outcomes.

The necessity analysis in this study reveals that no single conditional variable is essential for determining nurses' high or low QOL, offering a broader perspective beyond traditional research that often emphasizes individual factors. Previous studies have highlighted that nurses' QOL is significantly impacted by occupational stress, and burnout [41,42]. Additionally, social support has been consistently acknowledged as a protective element that enhances QOL [24]. Although these variables are crucial, this study reveals that their influence hinges on specific combinations of conditions rather than their isolated effects. For example, while social support significantly contributes to higher QOL, its positive influence depends on lower levels of psychological distress and manageable occupational stress. This configurational approach provides a more intricate understanding of how these factors interrelate to shape nurses' QOL.

This study identified four distinct configuration paths leading to high quality of life (QOL) among nurses, highlighting the interplay of psychological, occupational, and social factors.These paths emphasize different combinations of factors that contribute to high QOL, illustrating that no single factor alone determines nurses' well-being. Among these, H1 had the highest raw coverage (0.544), indicating its prominence in explaining high QOL. This finding underscores the critical role of reducing workplace stressors and fostering supportive environments in promoting nurses' well-being. Previous research has highlighted the negative impact of emotional exhaustion and occupational stress on QOL [43]. Consequently, these findings suggest that targeted interventions aimed at reducing psychological and occupational stress, while simultaneously strengthening social support, may be associated with improved QOL among nurses. Similarly, Xiao et al. (2021) demonstrated the buffering effect of social support in mitigating stress and improving resilience among nurses [44]. However, unlike prior studies that focused on isolated variables, this study's use of fsQCA reveals the synergistic effects of multiple factors in improving QOL.This study lies in its identification of specific configurations, such as H1, which shows that achieving high QOL does not require addressing all stress-related factors simultaneously. Instead, a targeted approach focusing on reducing burnout and occupational stress combined with enhancing social support is sufficient to significantly improve QOL. This multifactorial perspective offers a deeper understanding of the interactions among these factors, providing actionable insights for tailored and efficient interventions. This analysis directs nursing managers to focus on mitigating burnout and occupational stress while strengthening social support systems to effectively enhance nurses' QOL.

Moreover, further analysis from the configuration path perspective highlights that low QOL among nurses requires the concurrent presence of low social support and two core elements: high psychological stress and high occupational stress. Necessity analysis revealed a consistency of 0.913 and coverage of 0.545, indicating that 91.3% of nurses with low QOL exhibited these three core conditions together, and among these nurses, 54.5% demonstrated low QOL. This constellation of variables constitutes a necessary condition for low QOL. Low social support exacerbates the detrimental effects of psychological and occupational stress by depriving nurses of emotional, informational, and instrumental resources needed to cope with workplace demands [45]. High psychological distress and occupational stress, when coupled with insufficient support, create a cycle of emotional exhaustion and reduced resilience, further degrading QOL [43]. Consequently, nursing managers should focus on targeted interventions to reduce psychological and occupational stress while strengthening social support systems. Initiatives could include implementing stress reduction programs, offering access to counseling and mental health resources, and introducing workload management strategies. Moreover, fostering a supportive workplace culture through mentorship, peer support networks, and leadership recognition can provide nurses with the necessary resources to cope with stress. By addressing these core conditions, organizations can alleviate the factors contributing to low QOL and create a more sustainable and supportive environment for their nursing workforce.

It is also important to note that the configurations identified in this study are embedded within a specific organizational and cultural context. Tertiary hospitals in urban China operate under distinct structural conditions, including relatively high patient-to-nurse ratios, hierarchical management norms, and specific national healthcare staffing policies, all of which may shape how occupational stress, burnout, and social support interact to influence QOL. Future research should incorporate organizational culture and healthcare policy variables as additional conditions in QCA frameworks to better capture these contextual dynamics.

## 5. Limitation

This study has several limitations that should be considered when interpreting the findings. First, the cross-sectional design precludes any causal inference. Although fsQCA identifies sufficient configurations associated with QOL outcomes, the directionality and underlying mechanisms of these relationships cannot be established from this design alone. Longitudinal studies are needed to examine how configurations evolve over time. Second, the use of convenience sampling from two tertiary hospitals in urban Zhejiang Province, China, limits the representativeness and generalizability of the findings. Nurses working in primary care, rural hospitals, or different national healthcare systems may face substantially different occupational and social conditions. Future research should employ stratified or random sampling across diverse healthcare settings. Third, all variables were assessed via self-reported questionnaires, which are susceptible to social desirability bias and recall bias. These biases may have led to under- or over-reporting of distress, burnout, or social support levels. Future studies should consider integrating objective measures or triangulating findings with qualitative interview data. Fourth, the 2-item MBI used in this study does not capture personal accomplishment, a core burnout dimension, which may have resulted in an incomplete assessment of burnout's configurational role. Future studies are encouraged to employ the full Maslach Burnout Inventory. Fifth, fsQCA calibration requires researchers to specify membership thresholds, which involves a degree of subjectivity. Although calibration in this study was grounded in the theoretical ranges of each instrument and empirical distributions, sensitivity analyses using alternative calibration values are recommended. Finally, cultural and contextual factors specific to the Chinese healthcare system may influence the generalizability of the identified configurations to other cultural contexts. The study also did not include measures of organizational culture, institutional policies, or regional healthcare resource differences, which may serve as important contextual moderators.

## 6. Conclusions

This study utilized fuzzy-set Qualitative Comparative Analysis (fsQCA) to investigate the configurational relationships among psychological distress, occupational stress, social support, burnout, and work-family conflict and their impact on

nurses' QOL. As one of the first studies to apply fsQCA to examine both high and low QOL outcomes among nurses, it highlights the complex interplay of multiple factors rather than a single determinant. The analysis identified four configuration paths leading to high QOL and two paths associated with low QOL. The findings reaffirmed the significant roles of psychological distress, occupational stress, social support, burnout, and work-family conflict in shaping nurses' QOL. Notably, low occupational stress, low burnout, minimal psychological distress, and strong social and organizational support were key to achieving high QOL [46]. Conversely, high occupational stress and psychological distress, coupled with high work-family conflict and low social support, were central to low QOL. These insights emphasize the need for tailored interventions focusing on reducing occupational stress, mitigating work-family conflict, enhancing social support, and addressing psychological distress to improve nurses' well-being and overall quality of life. Organizational policies and management strategies should align with the identified configuration paths to create supportive and sustainable work environments for nurses.

## Author contributions

**Conceptualization:** Mao Dandan, Xu Jun.

**Data curation:** Mao Dandan, Xu Jun.

**Formal analysis:** Mao Dandan, Li Shasha.

**Methodology:** Mao Dandan, Li Shasha.

**Writing – original draft:** Mao Dandan, Li Shasha, Xu Jun.

**Writing – review & editing:** Mao Dandan, Xu Jun.

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
