## [Decision Letter · Decision Letter 0]

25 Feb 2026

PONE-D-25-65140Unraveling the Complexity of Nurses' Quality of Life: A QCA Approach to the Interplay of Psychological, Occupational, and Social FactorsPLOS One

Dear Dr. Jun,

Thank you for submitting your manuscript to PLOS ONE. After careful consideration, we feel that it has merit but does not fully meet PLOS ONE’s publication criteria as it currently stands. Therefore, we invite you to submit a revised version of the manuscript that addresses the points raised during the review process.

**ACADEMIC EDITOR:**

We look forward to receiving your revised manuscript.

Kind regards,

Omnia S. El Seifi, M.D., Ph.D.

Academic Editor

PLOS One

Journal Requirements:

2. "We noticed you have some minor occurrence of overlapping text with the following previous publication(s), which needs to be addressed:

https://pubmed.ncbi.nlm.nih.gov/38825685/

https://www.mdpi.com/1424-8220/24/24/7921

In your revision ensure you cite all your sources (including your own works), and quote or rephrase any duplicated text outside the methods section. Further consideration is dependent on these concerns being addressed.

3. In the online submission form you indicate that your data is not available for proprietary reasons and have provided a contact point for accessing this data. Please note that your current contact point is a co-author on this manuscript. According to our Data Policy, the contact point must not be an author on the manuscript and must be an institutional contact, ideally not an individual. Please revise your data statement to a non-author institutional point of contact, such as a data access or ethics committee, and send this to us via return email. Please also include contact information for the third party organization, and please include the full citation of where the data can be found.

6. Thank you for providing your underlying data as Supporting Information.

We note that the data set contains text or data that is not in English. Please note that PLOS is an English-language publisher, so we require data sets to be provided in English as well. Please upload an English-language version of your data set.

This will also allow us to determine if your data follows PLOS standards per our Data Availability policy here: https://journals.plos.org/plosone/s/data-availability.

Reviewers' comments:

Reviewer's Responses to Questions

**Comments to the Author**

1. Is the manuscript technically sound, and do the data support the conclusions?

Reviewer #1: Yes

Reviewer #2: Partly

2. Has the statistical analysis been performed appropriately and rigorously?

Reviewer #1: Yes

Reviewer #2: Yes

3. Have the authors made all data underlying the findings in their manuscript fully available?

Reviewer #1: Yes

Reviewer #2: Yes

4. Is the manuscript presented in an intelligible fashion and written in standard English?

Reviewer #1: Yes

Reviewer #2: No

5. Review Comments to the Author

Reviewer #1: language was clear, correct and unambiguous. except that they should consider writing in full the abbreviated word ‘QCA’ in the paper title. All other comments are written in the attached reviewer document

Reviewer #2: Regarding the introduction, it is extensive and up-to-date literature review clear identification of research gaps and innovative methodological approach (QCA) was used. However, several ideas are reiterated multiple times (e.g., occupational stress, burnout, psychological distress, work-family conflict). You may condense overlapping concepts and avoid repeating definitions. Also, the introduction mentions Work-Family Conflict Theory but does not integrate a broader theoretical framework (e.g., Conservation of Resources Theory, Job Demands-Resources Model). A guiding theory to explain how stressors and resources influence QOL may be added. There are minor inconsistencies and typographical issues: as spacing errors (e.g., “life satisfaction(Sili et al., 2022)”), inconsistent author formatting (Antolí -Jover) and some references appear outdated (e.g., 2012, 2015) without justification for historical relevance. Kindly, ensure APA/Harvard consistency and remove typographical errors.

As for the methods, using large sample size (n = 503) enhances statistical power and configurational diversity for fsQCA. Also, the clearly defined inclusion and exclusion criteria improve internal validity. However, the study lacks information on the country/region, hospital characteristics, and healthcare system context, which affects generalizability. Acknowledge selection bias in limitations. Also, there is no mention of response rate, refusal rate, or comparison between respondents and non-respondents. To improve interpretability, explain scoring and justify using SF-8 instead of profession-specific QOL tools. Kindly, justify using MBI-2 instead of full MBI or acknowledge reduced dimensionality because the 2-item version measures only emotional exhaustion and depersonalization, omitting personal accomplishment, which is a core burnout dimension. Most importantly, report ethics approval number and clarify data collection period. Moreover, the methods do not describe calibration procedures (anchor points for fuzzy sets), which is critical for fsQCA transparency, so describe calibration thresholds (full membership, crossover, non-membership). Minor grammatical and typographical errors (e.g., “Data analyze”) and inconsistent referencing style should be attended.

As for the results, key statistics (means, SDs, p-values, correlations, β coefficients, R², consistency, and coverage) are clearly reported, enhancing transparency and reproducibility. The identification of multiple sufficient configurations for high and low QOL reflects methodological novelty. However, the table contains formatting and labeling errors (e.g., unclear “Job position Yes/No”, missing N for working years less than 5 years. Also, the use of χ² (X²) for continuous QOL comparisons is unclear; ANOVA F-values would be more appropriate. The statement “social support was negatively correlated with QOL (r = 0.615)” is contradictory. A positive r indicates a positive correlation. This is a major reporting error that undermines credibility. Kindlyn review this interpretation. The statement “social support was negatively correlated with QOL (r = 0.615)” is contradictory. Regression assumptions better be reported (normality, multicollinearity diagnostics, VIF, tolerance). The results use causal language (“sufficient conditions,” “determine QOL”) despite cross-sectional data. This should be framed as configurational associations, not causal inference. Kindly, rephrase these sentences.

The discussion exhibits a strong interpretation of configurational findings, literature integration, and practical relevance. However, Overstated causal language should be reviewed. The discussion assumes interventions based on cross-sectional findings without acknowledging contextual constraints can be generalized (e.g., hospital type, cultural factors, healthcare system differences). This limits external validity. The discussion does not acknowledge major limitations (convenience sampling, self-report bias, CMV, cross-sectional design, cultural context). This is a critical omission for high-impact publications. Regarding the limitations, Transparent acknowledgment of sampling and self-report limitations; clear summary of configurational findings and strong practical relevance. The limitations are brief and not comprehensive. Critical issues are omitted, including, cross-sectional design limiting causal inference cultural/contextual specificity affecting external validity and fsQCA calibration and threshold subjectivity. The conclusion summarizes empirical findings but does not link them to theoretical frameworks. Future research suggestions are generic. More advanced directions (e.g., longitudinal QCA, mixed-method designs, intervention studies, cross-cultural comparisons) would strengthen scholarly contribution.

6. PLOS authors have the option to publish the peer review history of their article (what does this mean?). If published, this will include your full peer review and any attached files.

Reviewer #1: No

Reviewer #2: **Yes:** Dr. Lareen Magdi El-Sayed Abo-Seif

---

## [Author Response · Author response to Decision Letter 1]

12 Mar 2026

Dear Editor and Reviewers,

We sincerely thank the editor and both reviewers for their constructive and insightful comments. We have carefully addressed each point and made corresponding revisions to the manuscript. All changes are highlighted in the tracked-changes version. Below we provide a point-by-point response.

Response to Journal Requirements (Editor)

Comment 1: Manuscript format should comply with PLOS ONE style templates.

Response: We thank the editor for this reminder. The manuscript has been reformatted in accordance with PLOS ONE's style guidelines

Comment 2: Minor text overlap with two previously published works was identified.

Response: We thank the editor for identifying these overlaps. We have carefully reviewed the flagged passages, added appropriate citations to both works, and rewritten the overlapping sentences in our own words throughout the manuscript. The revised sections are highlighted in the tracked-changes version.

Comment 3: The data contact point must not be an author on the manuscript.

Response: We apologize for this oversight. The data availability statement has been revised. Data cannot be shared publicly because of participant privacy protections and institutional data governance regulations. Data are available from the Ningbo Municipal Hospital of Traditional Chinese Medicine (TCM), Affiliated Hospital of Zhejiang Chinese Medical University Institutional Data Access / Ethics Committee (contact via nbhospital@126.com) for researchers who meet the criteria for access to confidential data.

Comment 4: ORCID iD for the corresponding author must be validated.

Response: The corresponding author has logged into Editorial Manager and validated their ORCID iD. The ORCID is now linked and verified in the submission system.

Comment 5: Ethics statement should only appear in the Methods section.

Response: The ethics statement has been moved exclusively to the Methods section.

Comment 6: The uploaded dataset contains non-English content.

Response: We have prepared and uploaded an English-language version of the dataset. Variable names, labels, and value codes have all been translated and are consistent with the manuscript's terminology.

Response to Reviewer #1

Comment 1: The authors should consider writing the abbreviated word 'QCA' in full in the paper title.

Response: We thank for this suggestion. The title has been revised to include the full term.

Comment 2:Generalizability is limited due to convenience sampling and focus on two tertiary hospitals.

Response: We fully acknowledge this limitation. The Limitations section has been expanded to explicitly address this concern.

Comment 3:Self-reported data may introduce response bias (social desirability, recall bias).

Response: We agree that self-report methodology is a meaningful limitation. We have explicitly acknowledged response bias.

Comment 4:The study does not account for contextual factors such as organizational culture, healthcare policies, or regional differences.

Response: Thank you so much for this valuable suggestion. We have added acknowledgment of this limitation in the Limitations section and have incorporated a brief discussion in the Discussion section.

Comment 5:Data availability relies on request to corresponding author, limiting transparency.

Response: We thank the reviewer for raising this point. In accordance with both this recommendation and the journal's data policy requirement, we have updated the data availability statement.

At last,We greatly appreciate your careful review and valuable guidance, which has significantly improved the clarity and rigor of our manuscript.

Response to Reviewer #2

Comment 1: Several ideas are reiterated multiple times (e.g., occupational stress, burnout, psychological distress, work-family conflict). Overlapping concepts should be condensed.

Response: Thank you so much for this valuable suggestion. The introduction has been substantially revised to eliminate repetition. Each construct (occupational stress, psychological distress, burnout, work-family conflict, social support) is now introduced once with a concise definition and its relationship to QOL, followed by a synthesizing paragraph. Please see the revised Introduction.

Comment 2: The introduction mentions Work-Family Conflict Theory but does not integrate a broader theoretical framework (e.g., Conservation of Resources Theory, Job Demands-Resources Model).

Response: We thank the reviewer for this valuable suggestion. We have added a theoretical framework paragraph to the introduction, integrating the Conservation of Resources (COR) Theory (Hobfoll, 1989) and the Job Demands-Resources (JD-R) Model (Bakker & Demerouti, 2007). These frameworks help explain how stressors deplete personal and professional resources, while social support serves as a key resource that buffers against adverse QOL outcomes. The added paragraph is marked in the revised manuscript.

Comment 3: Minor inconsistencies and typographical issues: spacing errors, inconsistent author formatting, and some references appear outdated without justification.

Response: We have carefully proofread the entire manuscript. All spacing errors (e.g., "life satisfaction(Sili et al., 2022)" → "life satisfaction (Sili et al., 2022)") have been corrected. Author name formatting has been standardized (e.g., "Antolí-Jover" corrected throughout). References from 2012 and 2015 that were cited for foundational or historical definitions have been retained with brief contextual justification added in-text. All citations have been verified for APA consistency.

Comment 4: The study lacks information on the country/region, hospital characteristics, and healthcare system context.

Response: We appreciate this comment. We have added the following information to the Participants section: this study was conducted in Zhejiang, China. Both hospitals are tertiary-level general hospitals located in urban areas, serving a large and diverse patient population. This context is important for understanding the occupational demands faced by the nurses in our sample.

Comment 5: There is no mention of response rate, refusal rate, or comparison between respondents and non-respondents.

Response: We have added the following information to the Data Collection section: A total of 537 questionnaires were distributed, of which 520 were returned. After excluding 17 responses due to completion time under 3 minutes or patterned response styles, 503 valid responses were retained, yielding a valid response rate of 96.73%.

Comment 6: SF-8 selection should be justified instead of profession-specific QOL tools.

Response: We have added a justification sentence in the Measurements section: "The SF-8 was selected over profession-specific QOL instruments because it enables comparison with general population norms and across healthcare professions. Its brevity reduces respondent burden in busy clinical settings, and its validated Chinese version (Lang et al., 2018) ensures measurement reliability in our sample."

Comment 7: Justify using MBI-2 instead of full MBI or acknowledge reduced dimensionality.

Response: We acknowledge this limitation. We have added a justification in the Measurements section: "The MBI-2 was selected for its efficiency in rapid screening in large-scale surveys among healthcare workers (Li-Sauerwine et al., 2020). We acknowledge that, as a 2-item measure, it captures only emotional exhaustion and depersonalization and omits personal accomplishment, which is a core burnout dimension in the full MBI. This limitation is acknowledged in the Limitations section, and future studies are encouraged to employ the full MBI for a more comprehensive burnout assessment."

Comment 8: The methods do not describe calibration procedures (anchor points for fuzzy sets), which is critical for fsQCA transparency.

Response: Thank you so much for this valuable suggestion. We have added a detailed calibration paragraph to the Data Analysis section. Calibration was performed using the direct method in fsQCA 3.0. For each condition, three anchor points were defined: full membership (0.95), the crossover point (0.50), and full non-membership (0.05). The specific thresholds were determined based on theoretical meaning and the empirical distribution of each variable.

Comment 9: Grammatical errors (e.g., "Data analyze") and inconsistent referencing style.

Response: We are so sorry for that. "Data analyze" has been corrected to "Data Analysis". A full proofreading has been conducted and all grammatical and referencing inconsistencies have been corrected.

Comment 10: Ethics approval number should be reported; data collection period should be clarified.

Response: The ethics approval number ([Ethics Approval No.KYSL-2024-001-059]) has been added to the Ethics Statement in the Methods section. The data collection period has also been specified in the Data Collection subsection.

Comment 11: Table contains formatting and labeling errors (unclear "Job position Yes/No", missing N for working years ≤5 years).

Response: We are so sorry for that.. "Job position Yes/No" has been relabeled as "Management Position (Yes/No)" to improve clarity. The missing sample size for the "≤5 years" working experience group (N=113) has been restored. All table formatting has been reviewed and corrected.

Comment 12: The use of χ² for continuous QOL comparisons is unclear; ANOVA F-values would be more appropriate.

Response: We thank the reviewer for this observation. We confirm that one-way ANOVA was used for all group comparisons of QOL scores across demographic subgroups (as stated in the Data Analysis section). The column header in Table 1 has been corrected from "X²" to "F" to accurately reflect the statistical test used. We apologize for this labeling inconsistency.

Comment 13: "Social support was negatively correlated with QOL (r = 0.615)" is contradictory — a positive r value indicates a positive correlation.

Response: We sincerely apologize for this critical reporting error. This was a typographical error in the text. The Pearson correlation between social support and QOL is r = 0.615 (p < 0.01), which correctly indicates a positive correlation — higher social support is associated with higher QOL. The sentence has been corrected to read: "social support was positively correlated with QOL (r = 0.615, p < 0.01)." We have carefully re-examined all other correlation statements to ensure consistency with the reported r values.

Comment 14: Regression assumptions should be reported (normality, multicollinearity diagnostics, VIF, tolerance).

Response: We have added a paragraph reporting regression assumption checks: "Prior to hierarchical regression, assumptions were verified. Normality of residuals was confirmed via inspection of P-P plots and the Shapiro-Wilk test (p > 0.05). Multicollinearity was assessed using Variance Inflation Factor (VIF) and tolerance values. All VIF values were below 5 and tolerance values exceeded 0.2, indicating no significant multicollinearity among predictors."

Comment 15: The results use causal language despite cross-sectional data.

Response: We have carefully revised all instances of causal language throughout the Results section. Phrases such as "sufficient conditions for determining QOL" and "determine QOL" have been replaced with configurational/associational language, e.g., "configurations associated with high/low QOL" and "configurations sufficient for the outcome of high QOL."

Comment 16: Overstated causal language in the Discussion.

Response: All causal language in the Discussion has been revised. For example, "addressing these factors will reduce burnout" has been changed to "addressing these factors may be associated with reduced burnout." We have consistently framed our findings as configurational associations derived from cross-sectional data throughout the Discussion.

Comment 17: Discussion does not acknowledge major limitations.

Response: Thank you so much for the reminder. The Limitations section has been substantially expanded. Please see revised manuscript.

At last,We greatly appreciate your careful review and valuable guidance, which has significantly improved the clarity and rigor of our manuscript.

---

## [Decision Letter · Decision Letter 1]

20 Apr 2026

Unraveling the Complexity of Nurses' Quality of Life: A Fuzzy-Set Qualitative Comparative Analysis (fsQCA) Approach to the Interplay of Psychological, Occupational, and Social Factors

PONE-D-25-65140R1

Dear Authors

We’re pleased to inform you that your manuscript has been judged scientifically suitable for publication and will be formally accepted for publication once it meets all outstanding technical requirements.

Kind regards,

Ahmed Abdelwahab Ibrahim El-Sayed,

Academic Editor

PLOS One

Additional Editor Comments (optional):

Dear Authors,

Thank you for submitting your study. I am pleased to inform you that your manuscript is accepted for publication in PLOS ONE in its current form.

Congratulations on this achievement.

Reviewers' comments:

Reviewer's Responses to Questions

**Comments to the Author**

1. If the authors have adequately addressed your comments raised in a previous round of review and you feel that this manuscript is now acceptable for publication, you may indicate that here to bypass the “Comments to the Author” section, enter your conflict of interest statement in the “Confidential to Editor” section, and submit your "Accept" recommendation.

Reviewer #1: All comments have been addressed

Reviewer #2: All comments have been addressed

2. Is the manuscript technically sound, and do the data support the conclusions?

Reviewer #1: Yes

Reviewer #2: Yes

3. Has the statistical analysis been performed appropriately and rigorously?

Reviewer #1: Yes

Reviewer #2: Yes

4. Have the authors made all data underlying the findings in their manuscript fully available?

Reviewer #1: Yes

Reviewer #2: Yes

5. Is the manuscript presented in an intelligible fashion and written in standard English?

Reviewer #1: Yes

Reviewer #2: Yes

6. Review Comments to the Author

Reviewer #1: I have reviewed the paper and seen that all comments raised by reviewers have been duly addressed. authors have also explained why they can not make some part of the data available to reviewers. According to their submissions, The data availability statement has been revised. Data cannot be shared publicly because of participant privacy protections and institutional data governance regulations. Data are available from the Ningbo Municipal

Hospital of Traditional Chinese Medicine (TCM), Affiliated Hospital of Zhejiang Chinese

Medical University Institutional Data Access / Ethics Committee (contact via

nbhospital@126.com) for researchers who meet the criteria for access to confidential

data.

I therefore recommend it for publication since it meets the guidelines of the journal.

Reviewer #2: (No Response)

7. PLOS authors have the option to publish the peer review history of their article (what does this mean?). If published, this will include your full peer review and any attached files.

Reviewer #1: **Yes:** DORIS HAGAN

Reviewer #2: **Yes:** Dr. Lareen Magdi El-Sayed Abo-Seif

---

## [Editor Report · Acceptance letter]

PONE-D-25-65140R1

PLOS One

Dear Dr. Dandan,

I'm pleased to inform you that your manuscript has been deemed suitable for publication in PLOS One. Congratulations! Your manuscript is now being handed over to our production team.

Kind regards,

on behalf of

Dr. Ahmed Abdelwahab Ibrahim El-Sayed

Academic Editor

PLOS One